# Bridge ties bind collective memories

Ida Momennejad[1,2,3], Ajua Duker[1,4] & Alin Coman[1]

From families to nations, what binds individuals in social groups is, to a large degree, their shared beliefs, norms, and memories. These emergent outcomes are thought to occur because communication among individuals results in community-wide synchronization. Here, we use experimental manipulations in lab-created networks to investigate how the temporal dynamics of conversations shape the formation of collective memories. We show that when individuals that bridge between clusters (i.e., bridge ties) communicate early on in a series of networked interactions, the network reaches higher mnemonic convergence compared to when individuals first interact within clusters (i.e., cluster ties). This effect, we show, is due to the tradeoffs between initial information diversity and accumulated overlap over time. Our approach provides a framework to analyze and design interventions in social networks that optimize information sharing and diminish the likelihood of information bubbles and polarization.

[1] Department of Psychology, Princeton University, Princeton, NJ 08544, USA. [2] Princeton Neuroscience Institute, Princeton University, Princeton, NJ 08544, USA. [3] Present address: Department of Bioengineering, Columbia University, New York 10027, USA. [4] Present address: Department of Psychology, Yale University, New Haven, CT 06520-8205, USA. Correspondence and requests for materials should be addressed to I.M. (email: idam@princeton.edu) or to A.C. (email: acoman@princeton.edu)

Social interactions are crucial to communities that engage in coordinated behavior. These interactions constitute the means through which beliefs, memories, and norms become shared across communities. They can facilitate the spread of information about healthy behaviors[1], change negative norms[2], and enable large-scale cooperation[3]. On the other hand, interactions within homophilous social clusters[4] give rise to information bubbles[5] and political polarization[6,7], and have the potential to disrupt optimal collective behavior[8,9]. Exploring the impact of social interactions on large-scale phenomena has recently led to significant advances in understanding the formation of collective memories[10,11]. Despite these advances, we know very little about the dynamical processes involved in the community-wide synchronization of memories (i.e., collective memories). Here, we show experimentally that the temporal sequence of conversations in social networks impacts the degree to which communities converge on a shared memory of the past. This investigation has the potential to illuminate phenomena that are directly dependent on a network's collective memories, such as the mobilization of collective identity[6,7] and collective behavior[8,9], and to highlight how the interplay between micro-level cognitive processes and the structural and temporal features of social networks can give rise to large-scale social phenomena[1,12].

We build on the extensive psychological research showing that once an event is encoded by an individual, its memory is malleable[13–15]. It is subject to cognitive transformations, such as forgetting and distortion[16], and susceptible to social influences[17,18]. Due to this malleability, conversationally remembering the past often leads to the synchronization of memories between interacting partners[19]. When these dyadic-level influences are part of a larger network of social interactions, collective memories emerge[10,17,20,21]. In order to understand the community-level synchronization of memories, current theoretical models point to the need to develop a framework that captures both how an individual's memories are shaped in social interactions, as well as how features of the social network that characterize the community's interactions impact the formation of collective memories[22–24]. Since collective memories are dependent on the cognitive operations of the individuals who comprise the community, we aim to explore how repeated recollections in a social setting affect people's memories of an experienced event[25]. At a network level, we investigate how the temporal sequencing of conversations in the social network affects the degree of convergence that the community reaches. Imagine a situation in which a community of individuals experiences a public event through mass-media exposure (e.g., the September 11 attacks). They then start communicating with one another about the information they acquired about the event. We are interested here in understanding how convergent the community's memories become following these conversations and how this convergence is influenced by the temporal nature that characterizes the community's interactions.

Previous research has found that the influence that one individual exerts over another can propagate through the network and impacts the degree to which communities converge on a similar memory of an experienced event. This research has shown, for instance, that networks characterized by clusters that are highly connected with one another form more convergent collective memories than networks comprised of sparsely connected clusters. This is because connections between clusters allow for information to propagate through the network, which synchronizes the community members' memories[10]. Not all individual members are, however, equally influential in their potential to affect the collective memory of the larger network. Individuals who connect between clusters (i.e., bridge ties) have a

significant influence in the network[26,27]. No research to date has experimentally explored how these ties facilitate the formation of collective memories across a social network, a gap that we intend to address herein.

Crucially, social interactions within communities unfold over time. Depending on the sequential order of conversations, a "bridge tie" may never get the chance to impact the network, especially if it occurs after the community had already engaged in extensive interactions in isolated clusters. Most previous investigations use static topological mappings to showcase the impact of bridge ties[26]. In contrast with these approaches, we use a temporal network framework to understand when "bridge tie" conversations should take place to maximally impact the convergence of memories across the community[28,29]. To do so, we experimentally manipulate the temporal order of "bridge tie" and "cluster tie" conversations in lab-created networks. We then measure how this manipulation impacts the formation of collective memories.

Our hypothesis is that if participants who are connected through a bridge tie discuss memories of a commonly experienced event early on, they will facilitate widespread mnemonic convergence in the network. This is because the dyadic-level synchronization between the individuals who bridge between clusters will influence the subsequent conversations within the clusters. In contrast, early alignment between individuals within each cluster (i.e., cluster ties) should lead to less mnemonic convergence across the community. This is because conversations among individuals within clusters continuously reinforce their cluster's memories in a way that makes these memories less sensitive to influences from neighboring clusters in subsequent conversations across the clusters. To test this hypothesis, we conducted a laboratory experiment in which we kept the topological properties of conversational networks constant across experimental conditions (i.e., all nodes have the same degree, closeness centrality, betweenness centrality, and eigenvector centrality) and only manipulate the temporal order of conversations within these networks (i.e., link order). This temporal order is manipulated such that the first round of conversations occurs either on bridge ties (the Bridge Ties First condition) or on cluster ties (the Cluster Ties First condition) (Fig. 1).

One hundred and ninety-two participants enrolled in the study through Princeton University's recruitment system. They were assigned to 16-member communities, here defined as clusters of interconnected individuals within a social network (Fig. 1). All participants completed the experimental procedure on lab computers. In the *study phase* (phase 1), participants read a story that contained 30 critical items[16]. Then, in the *pre-conversational recall phase* (phase 2), they individually recalled the studied information. In the *conversational recall phase* (phase 3), each participant in the 16-member network engaged in a series of four anonymous dyadic conversations (each with a different partner), during which they were instructed to jointly remember the studied materials. Conversations took the form of interactive exchanges in a chat-like, computer-mediated environment in which participants typed their recollections. Finally, in the *post-conversational recall phase* (phase 4), they individually recalled the initially studied information once again (Fig. 2).

In the conversational recall phase, each participant engaged in a sequence of four 150 s conversations. In the Bridge Ties First condition ($n = 96$ participants; six 16-member networks), the conversational sequence began with interactions between individuals who belonged to different pre-determined clusters (i.e., bridge tie). In the Cluster Ties First condition ($n = 96$ participants; six 16-member networks), the first conversation occurred between individuals who were part of the same pre-determined cluster (i.e., cluster tie). The second and third conversations took

place within clusters in both conditions (cluster ties), while the fourth conversation again differentiated between the Bridge Ties First condition, in which participants now communicated within the cluster, and the Cluster Ties First condition, in which participants communicated between clusters (Figs. 1 and 2).

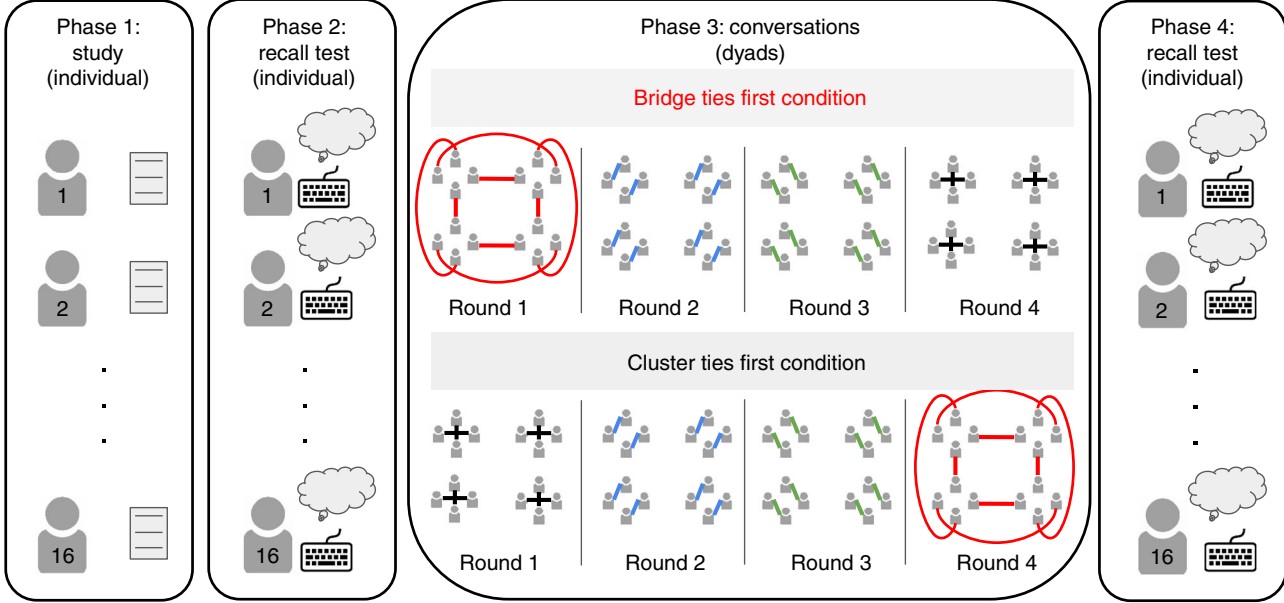

**Fig. 1** A graphic summary of bridge ties (red), cluster ties (black), and clusters. For each cluster, its neighboring and distant clusters are shown. Mnemonic convergence is measured across the entire 16-member network. Average mnemonic similarity is measured (1) within cluster (among the four members of that cluster), (2) between a cluster and its neighboring cluster (e.g., cluster a and cluster b), and (3) between a cluster and its distant neighbor cluster (e.g., cluster a and cluster c)

Participants were assigned to their position in clusters randomly and did not have knowledge of the structure of the network.

For our dependent measures, we computed scores that quantitatively captured the formation of collective memory in the community. We refer to these scores as mnemonic convergence scores when they involve the entire community (of 16 participants) and mnemonic similarity scores when they involve only sub-sections of the community. To compute these scores, we adapted the procedure established by Coman et al.[10]. Each person's memory was individually measured once before the conversational rounds started and once after all conversational rounds were completed. At each of these two time-points, a participant's memory was measured in terms of free recall of 30 items from the story they had read. The recalls were operationalized as a vector with 30 slots corresponding to the 30 studied items. For each element of the memory vector, a value of 1 indicated that an item was recalled and 0 indicated that the item was not recalled (see Supplementary Methods). Using these vectors, we first calculated a mnemonic similarity score for each pair of participants within a network by dividing the number of items the two participants remembered in common by the total number of items (of the 30) that either participant in the pair remembered[11]. As a hypothetical example, if participant A remembered items 1 and 2 and participant B remembered items 2 and 4 from a 4-item stimulus set, then their mnemonic similarity score is 0.33, computed as the division of 1 (item remembered in common) by 3 (total items remembered by either participant in the pair). Using these pairwise mnemonic similarity scores, we computed: (a) a global-level mnemonic convergence score by averaging the mnemonic similarity scores across all pairs of participants in the network and (b) local-level average mnemonic similarity scores by averaging the mnemonic similarity scores depending on the positions of the participants in the network: (i) *within-cluster mnemonic similarity* by averaging the mnemonic similarity scores of participants who were part of the same cluster (e.g., participants in cluster a in Fig. 1), (ii) *neighboring-cluster mnemonic similarity* by averaging the mnemonic similarity scores

**Fig. 2** Phases of the experimental procedure. Each experimental session had four phases. In phase 1, all 16 participants that comprised a lab-created community studied the material (see Supplementary Methods) alone, and in phase 2 (as in phase 4), each participant engaged in individual free recall by typing their recollections on the computer. In phase 3, the nodes represent 16 participants and the edges represent conversations between two individuals. The order of the conversations in phase 3 depended on the condition: in the Bridge Ties First condition participants have their first conversation across clusters, whereas in the Cluster Ties First condition they have their first conversation within clusters. The Keyboard image is from http://icons8.com. All rights reserved

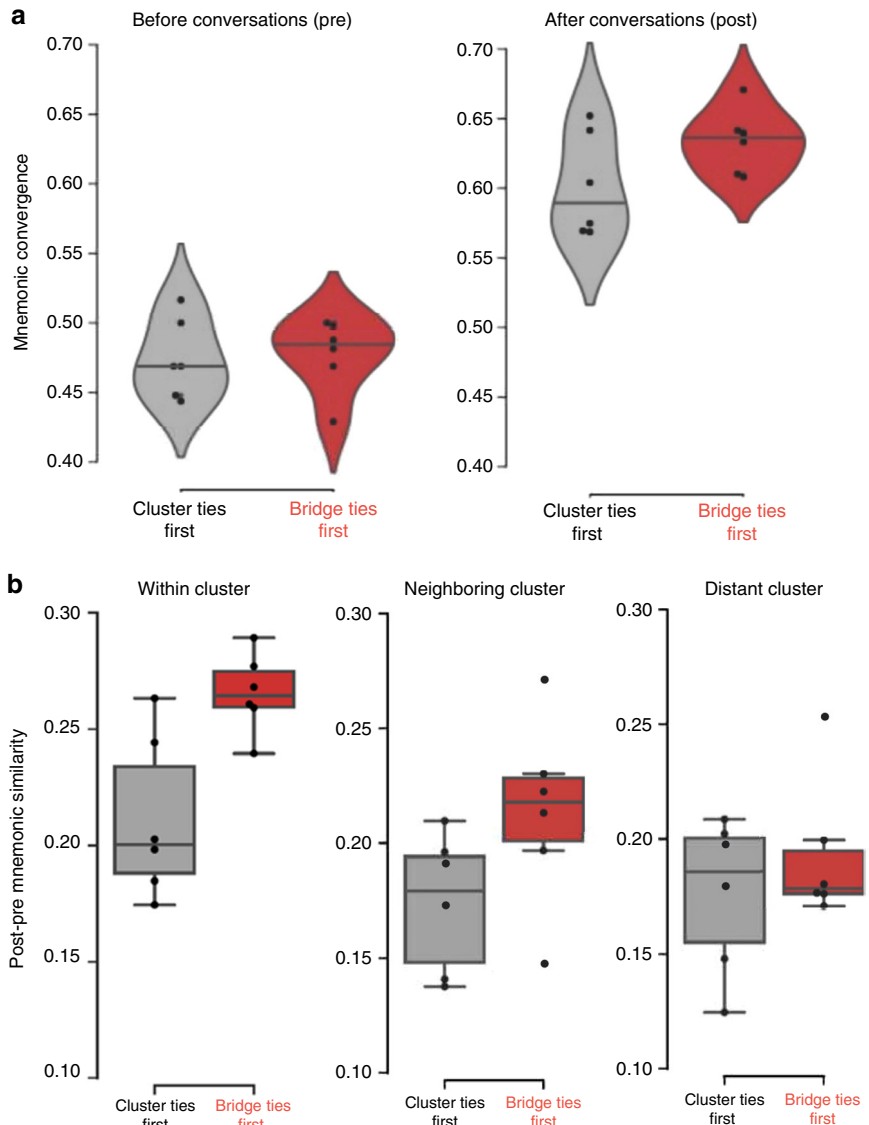

**Fig. 3** Mnemonic similarity and convergence scores. **a** The increase in mnemonic convergence from pre-conversation to post-conversation was larger in the Bridge Ties First condition (M = 0.22; SD = 0.03) than in the Cluster Ties First condition (M = 0.18; SD = 0.03), $t(10) = 2.21$, $p = 0.052$, Cohen's $d = 1.28$, confidence interval (CI) [0.00, 0.07]. **b** The increase in mnemonic similarity from pre- to post-conversation was larger in the Bridge Ties First condition than in the Cluster Ties First condition for participants who were part of the same cluster ($M_{BridgeTies} = 0.27$, $SD_{BridgeTies} = 0.02$ vs. $M_{ClusterTies} = 0.21$; $SD_{ClusterTies} = 0.03$; $t(10) = 3.43$, $p = 0.006$, Cohen's $d = 2.10$, CI [0.02, 0.09]) and marginally significant for participants in neighboring clusters ($M_{BridgeTies} = 0.21$, $SD_{BridgeTies} = 0.04$ vs. $M_{ClusterTies} = 0.17$; $SD_{ClusterTies} = 0.03$; $t(10) = 2.11$, $p = 0.061$, Cohen's $d = 1.11$, CI [0.00, 0.08]), but not for those in distant clusters ($p = 0.39$). All comparisons involve independent-sample $t$-tests. In this figure, boxplots show interquartile range (box), mean (black line within interquartile range), and data range (vertical lines)

of non-interacting participants who belonged to adjacent clusters (e.g., participants from cluster a and participants from cluster b in Fig. 1), and (iii) *distant-cluster mnemonic similarity* by averaging the mnemonic similarity scores of participants who belonged to non-adjacent clusters (e.g., participants from cluster a and participants from cluster c in Fig. 1). These measures were computed separately for the pre-conversational and post-conversational recalls. A mnemonic convergence (and mnemonic similarity) score of 0 indicates that participants remembered nothing in common, while a score of 1 indicates perfect overlap among participants.

## Results

**Dynamics of mnemonic convergence**. To explore whether the temporal sequence of conversations impacts the emergence of

collective memories, we first compared the mnemonic convergence scores in the two conditions. Consistent with our hypothesis, we found that the mnemonic convergence increased from pre- to post-conversation to a larger degree in the Bridge Ties First condition than in the Cluster Ties First condition (Fig. 3a). That is, Bridge Ties First communities reached more convergent collective memories than Cluster Ties First communities. This pattern, we argued, is due to the fact that bridge ties lead to the increased mnemonic similarity among individuals who belong to connected clusters over time. In order to investigate this claim we compared the neighboring-cluster similarity scores between the two conditions. We used a score of mnemonic similarity change by subtracting the average mnemonic similarity of the pre-conversational recalls from the post-conversational average mnemonic similarity scores.

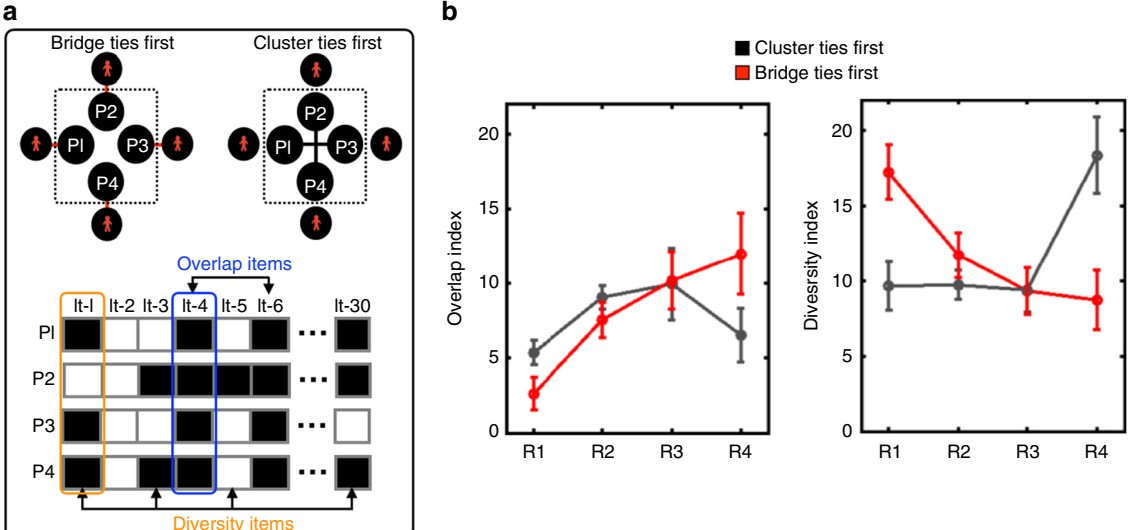

**Fig. 4** Memory overlap and memory diversity indices. **a** For each cluster (designated with the dotted line), we compared vectors corresponding to information that came up in conversations of each of its four members (participants P1 through P4). Some of these conversations were within cluster, and some were between a cluster member and an "outsider" (red figures). For each round, a cluster's overlap index indicated the number of items (out of the 30 studied items) that came up in conversations of all members (highlighted with the blue box), while the diversity index indicated the number of items that were mentioned in at least one but no more than three conversations (highlighted with the orange box). **b** The dynamics of the overlap and diversity indices, by round, separate for the Bridge Ties First (red) and Cluster Ties First (black) conditions. A repeated-measures analysis of variance (ANOVA) revealed that the interaction between Time and Condition is significant for both the overlap [$F(3, 8) = 11.76$, $\eta^2 = 0.81$, $p = 0.003$] and the diversity [$F(3, 8) = 31.53$, $\eta^2 = 0.92$, $p = 0.001$] indices, with post-hoc independent-sample $t$-tests indicating significant differences between the Bridge and Cluster Ties First conditions in both Round 1 (Overlap: $t(10) = 4.97$, $d = 2.89$, confidence interval (CI) [1.52, 3.98], $p < 0.001$; Diversity: $t(10) = 7.69$, $d = 4.46$, CI [5.39, 9.78], $p < 0.001$) and round 4 (Overlap: $t(10) = 4.08$, $d = 2.41$, CI [2.48, 8.43], $p < 0.002$; Diversity: $t(10) = 7.34$, $d = 4.28$, CI [6.71, 12.54], $p < 0.001$). Error bars represent standard deviation around the mean

We found support for our prediction that bridge ties affect network-wide mnemonic convergence by aligning the memories of individuals who are part of neighboring clusters. In the Bridge Ties First condition, participants' memories were more similar to members of their neighboring cluster than in the Cluster Ties First condition (Fig. 3b). We predicted no difference between the two conditions in distant-cluster similarity, because in both conditions the influence of one participant over another's memories can only propagate into the neighboring cluster and no further. Indeed, the mnemonic similarity change was not statistically different between the Bridge Ties First and the Cluster Ties first conditions for distant cluster comparisons (Fig. 3b).

**Differences in information diversity and accumulated overlap.** As to the alignment of memories among within-cluster participants, two possibilities emerge. The first possibility builds on the assumption that within-cluster participants in the Cluster Ties First condition might align their memories in the first round. They then continually rehearse these memories in subsequent rounds, thus forming locally convergent memories resistant to influence from neighboring clusters in the last round. This account would predict a larger increase in within-cluster mnemonic similarity in the Cluster Ties First condition than in the Bridge Ties First condition. An alternative possibility, consistent with the collaborative learning literature[30,31], is that this early consolidated alignment in the Cluster Ties First condition might be perturbed by the introduction of information from neighboring clusters in the final round. As each member of the cluster aligns with a different "outsider" in the final conversational round, the cluster may diverge from their locally formed collective memory in different directions. This latter account would predict less within-cluster average similarity in the Cluster Ties First compared to the Bridge Ties First condition. Indeed, we find

support for the latter hypothesis, with the Cluster Ties First condition reaching less within-cluster average similarity than the Bridge Ties First condition (Fig. 3).

As mentioned, we conjecture that this pattern of results is driven by the temporal dynamics of *information diversity* and *accumulated overlap* that the clusters experience during the different rounds of conversational remembering. To illustrate, imagine a four-member group that discusses (in sequential dyadic interactions) an event previously experienced by each member from their unique perspectives. Initially, the different members of the group might introduce in their discussions information that pertains to their unique perspectives (*high information diversity*). As subsequent conversations unfold, these unique items are integrated into their individual memories and thus become shared among all the members of the four-member group (*accumulated overlap*). We propose that the two conditions, Bridge Ties First and Cluster Ties First, exhibit different dynamics of *information diversity* and *accumulated overlap* as follows. In round 1 of the Bridge Ties First condition, each participant had a conversation with an individual from another cluster (see Fig. 4a). Thus, eight individuals contributed to the pool of items collectively recalled by the cluster in round 1. As a consequence, in this round, the pool of items that the cluster remembered collectively should be characterized by *high information diversity* (i.e., number of items remembered in at least one, but no more than three conversations of the participants in the cluster) and *low accumulated overlap* (i.e., number of items remembered in all conversations of participants who form a cluster). In contrast, in round 1 of the Cluster Ties First condition, the collective pool of information within a cluster should be characterized by *low information diversity* and *high accumulated overlap*, because their round 1 interactions occur within the cluster and only four individuals contribute to the cluster's collective pool of

information. In round 4, the opposite pattern should occur, with the Cluster Ties First condition experiencing an increase in information diversity (and decrease in accumulated overlap) due to the final interactions with participants outside the cluster, and the Bridge Ties First condition experiencing a decrease in information diversity (and increase in accumulated overlap) due to repeated interactions within the cluster.

To test this hypothesis, we computed two indices that characterize the information that is available to each cluster: an *information diversity index* and an *information overlap index*. For each participant within a cluster, and for each conversational round, we constructed a 30-item vector to capture the information produced in his/her conversation. Because each cluster involves four participants, there were four vectors that went into the computation, one for each participant. If an item (among the 30 initially studied) was present in all four vectors, it was designated as an overlap item. The total number of overlap items constituted the *information overlap index* for the cluster. If an item was present in at least one vector, but in no more than three vectors, it was designated as a diversity item (Fig. 4a). The total number of diversity items constituted the *information diversity index* for the cluster. These indices were computed for each cluster and then averaged across the four clusters in a network.

Consistent with our conjecture, we found that in the Bridge Ties First condition, the information was initially (round 1) less overlapping and more diverse than in the Cluster Ties First condition. As participants engaged in subsequent conversations, this dynamic reversed, such that information brought up in the last round was more diverse and had a lower overlap score in the Cluster Ties First condition than in the Bridge Ties First condition. This dynamic, we argue, resulted in higher within-cluster alignment in the Bridge Ties First condition, but had the opposite effect in the Cluster Ties First condition, where the information that was introduced in round 4 had no time to be integrated and discussed within the cluster (Fig. 4b). It is worth noting that the prediction derived from this explanation is that additional rounds of within-cluster conversations (i.e., round 5 and on) in both conditions should result in an increase in the overlap index and decrease in the diversity index, since information is continuously entered into the common pool of items that are conversationally reinforced.

## Discussion

In this paper we offer a framework for quantifying the temporal dynamics involved in the formation of collective memories in social networks. We have shown experimentally, for the first time, that individuals who bridge network clusters have a strong influence on the emerging collective memory in lab-created networks. The approach developed here provides a significant advance in our understanding of these emergent outcomes. We show that the same network topology can give rise to different convergent collective memories, depending on the temporal sequencing of conversations.

The literature on collective memory would benefit from systematic investigations of the impact of network characteristics on the dynamical transformation of collective memories[32]. We are only beginning to understand how the malleability of human memory impacts the formation of collective memories. There are a multitude of factors that could interact with the temporal and topological features of social networks to give rise to collective memories. At a psychological level, people's attitudes[33], the perceived similarity among interacting participants[34,35], and the medium in which interactions take place (online vs. face to face)[36] likely play a role in the synchronization dynamics involving social groups. For instance, in our current investigation participants

interacted with one another in computer-mediated interactions that involved turn-taking. Even though the experimental situation we created is representative of today's social media landscape (e.g., Twitter, Facebook), it is likely that face-to-face conversations in which social cues are much more salient would result in different outcomes. More research is needed, thus, into how face-to-face conversations might affect large-scale phenomena differently than computer-mediated interactions. Our conjecture is that the more social information the participants have of one another, the more pronounced the social influence processes will be. More specifically, similarity cues would facilitate social influence, while dissimilarity cues would diminish social influence, relative to control conditions in which little or no social information is provided.

Further investigations into how the topological and temporal features of social networks affect the formation of collective memories are certainly worthwhile. One such investigation could explore the critical bridge width (i.e., the length of the chain that bridges between two clusters) at which collective memories are still affected by the temporal sequencing of conversations. Indeed, in real-world communities, bridges among clusters are oftentimes of widths larger than 1[37]. At what width would one expect the influence of bridge ties to be eliminated? Previous studies suggest that the social influence from one individual to another does not spread more than 3 degrees away from the originating source[10,36], which might constitute the upper limit at which the critical bridge width can still affect mnemonic convergence. Another potential extension of the current work involves establishing whether social influences on memory involve simple contagions (one source necessary for memory change) or complex contagions (multiple sources necessary for memory change). The answer to this question could reveal the types of networks most conducive to the formation of highly convergent collective memories[1,10,12,38]. In this context, varying the number of bridge ties between clusters in larger networks would provide more nuanced answers to questions about the optimal connectivity among clusters that would facilitate the formation of collective memories.

Given observed topological and temporal features of the community's interactions, one could easily make predictions about how convergent their collective memories will become. Being able to make these predictions is of tremendous importance, given that the collective memory a network reaches could predict a host of large-scale phenomena, from information bubbles[5], to political polarization[6,7], and to collective behavior[8,9]. It is worth noting that in most real-world circumstances, communities are more likely to resemble the Cluster Ties First condition, with initial conversations taking place within clusters, and only subsequent conversations reaching outside the cluster. There are situations, however, when control could be exercised over the temporal sequence of social interactions (e.g., study groups in educational settings, sequencing of meetings in organizations, etc.). One could design these interactions to optimize for desired outcomes, such as mnemonic convergence, information homogeneity and diversity, and recall accuracy.

We focused on how communities come to hold a collective memory of an emotionally neutral event that all participants experienced. We made the decision to restrict the experimental situation to this setting in order to rigorously isolate the variables that are of interest to this investigation. In their day-to-day lives, however, people often encounter emotionally charged events, the propagation of which might be modulated by motivated reasoning processes. Based on previous research, we conjecture that emotion should facilitate the propagation of social influence in networks in a way that would increase community-wide convergence[39]. If one experiences an emotional reaction when witnessing an event, then one is more likely to relay it to another

individual, which in turn would facilitate convergence processes[40]. This propagation-convergence association would be influenced, however, by people's prior beliefs, memories, and motivations. That is, groups of similar-minded individuals might perceive an event through the prism of their identity in a way that makes them converge on a similar memory of the past. These motivational biases could lead rival groups to converge on different memories of the same event, as famously shown in previous research[32,33]. Oftentimes, these biases create a context for communities to converge on inaccurate information[41]. In the current investigation, the rate of memory distortion was very low, which limits the conclusions one could draw based on these data. However, systematic experimental manipulations involving (i) slightly different events across community members, (ii) ambiguity of the experienced event, and (iii) the different motivations of the community members will likely reveal meaningful dynamics involving the formation of false collective memories.

Ultimately, understanding collective cognition will clarify social phenomena that underlie significant challenges of our times: information bubbles. Highly clustered networks, in which individuals reinforce their memories in repeated within-cluster interactions, lead to fragmented collective memories. This, in turn, could produce information bubbles, largely because the different factions rarely exchange information to reach common ground[38]. Our proposed approach can help strategically optimize information ecosystems for maximal knowledge acquisition and efficient community organization. These predictions would provide the grounding for interventions to avoid information bubbles and to reduce the spread of misinformation in social networks.

## Methods

**Participants**. A total of 192 students (123 female, mean age 21.83 years, SD = 3.97) affiliated with Princeton University took part in the study voluntarily for either research credit or compensation. The stopping rule for participant recruitment was established based on the effect size obtained in a previous study that used aggregate measures of collective memory in networks[10]. A sample size of 12 networks was deemed adequate to obtain an effect size of Cohen's $d = 1$ for the planned between-condition comparisons. The relatively small sample size needed to reveal an effect is due to the fact that the aggregation procedure involves averaging over all pairwise scores within each network (i.e., 120 scores per network), which drastically reduces the standard deviation and standard error in each condition. No participants were excluded from analyses. They were grouped into twelve 16-member networks and went through the study in a Princeton computer lab, which contains visually partitioned computers. The participants interacted anonymously through the software SoPHIE (Software Platform for Human Interaction Experiments); they were from a wide range of fields of study, which made it unlikely that any subject would know more than one other person in the room. In total, 12 sessions were conducted, each involving a 16-person network in a between-network design: participants in 6 networks were in the Bridge Ties first condition (96 participants) and the remaining 6 networks were in the Cluster Ties first condition (96 participants). Assignment of network to condition was random. All subjects gave informed consent for the protocol, which was approved by Princeton University's Institutional Review Board.

**Stimuli**. Using the Qualtrics survey paradigm, we presented participants with a story taken from ref. [16]. The 30-item story contains information about two boys who skip school and visit one of the boy's house. Even though the story was initially designed to contain items relevant for two cognitive schemas (i.e., real-estate mindset vs. burglar mindset), our manipulation did not involve the activation of these schemas. We used this story because it has been widely employed in psychological investigations of memory in individual-level studies and, as a consequence, a well-established coding scheme has been already developed.

**Design and procedure**. Participants signed up for the study through Princeton University's online recruitment system. Each session was conducted with 16 participants who went through the experimental procedure together. Participants within each network were physically present in the same room and carried out the study on the designated computer terminals throughout. In the study phase, participants initially studied the story for a fixed amount of time. They were told that their memory will be tested in a later phase. Then, in a pre-conversational recall phase, participants were asked to individually remember as much as they could about the initially presented information. After this phase, participants took part in a sequence of conversations for which they were instructed to jointly

remember the initially studied materials (conversational recall). These computer-mediated chat conversations took place in dyads, such that each participant in a 16-member cluster engaged in a sequence of 4 interactions, each lasting for 150 s. Participants did not know the identity of their chat partners and merely saw an assigned avatar that indicated the uniqueness of their conversation partner in each round. The conversations were characterized by turn-taking, with virtually all conversational recall instances involving collaboration between the interacting partners. Of note, both participants who engaged in a conversation saw the conversation as it unfolded on the screen and were able to scroll to any point of the conversation they wished. A qualitative analysis of the conversations revealed that all of the participants stayed on task throughout the duration of the study and engaged in collaboratively remembering the initially studied materials, as instructed. Following the conversational recalls, participants were instructed to once again individually recall the initially studied story. For both the pre- and post-conversational recalls, the participants were given as much time as needed to type their recalls; the recall time ranged between 4 and 8 min across participants.

For the conversational recall phase, we manipulated the network structure of the conversational interactions as illustrated in Fig. 2. A software platform was specifically designed for this project to allow for fluent computer-mediated interactions among participants (i.e., SoPHIE). We kept the number of participants and the number of conversations in which each participant was engaged constant across the two conditions. The only difference between the two conditions was the sequencing of conversational recall sessions. In the Bridge Ties first condition (6 networks), the first conversation occurred between participants who were part of different clusters, while in the Cluster Ties first condition (6 networks), the first conversation took place between participants who were part of the same cluster. Each conversation lasted for 150 s; the preliminary studies showed that it provided ample time for information to be exchanged. A final recall test followed the conversational phase (post-conversational recall).

Coding of all of the recall protocols was performed by a research assistant who was blind to the study's hypotheses and involved a binary system in which an item was labeled as either remembered or not remembered. The coding scheme followed the designation employed in the original study by Anderson and Pichert[16] for 30 predefined memory items included in the story (see Supplementary Methods). For each item, a score of 1 indicated that it was remembered and 0 indicated that the item was not recalled. Items were coded for "gist", meaning that the participant did not need to recall each scoring unit verbatim in order for it to count. For example, although a scoring unit on the scale reads "The basement is damp", if a participant mentioned a "soggy cellar," the unit was counted as remembered. Due to this gist-based coding, we had very few recall distortions in the data (<1%) and we coded these distortions in a gist-based consistent manner, with recall units that contained egregious errors not being counted as recalled (e.g., "there was a TV" when in fact "there was a stereo") and minimal errors were counted as correctly remembered (e.g., "there was a computer" instead of the studied item: "there was a laptop"). As such, each participant's recall during each phase could be captured in a 30-item recall vector with 0 and 1 scores. Ten percent of the data were double-coded for reliability (Cohen's $\kappa = 0.84$). The double coding was performed by research assistants who were also blind to the hypotheses of the study. The 3- to 5-min distracter tasks, in which participants completed unrelated questionnaires, were inserted between any two phases described above.

**Reporting Summary**. Further information on experimental design is available in the Nature Research Reporting Summary linked to this article.

## Data availability
All data associated with this study is available on an open data platform [https://osf.io/fxky4/?view_only=ededdaf259e74523a59c08e9716aa025].

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

## Acknowledgements

We gratefully acknowledge the support of National Science Foundation Grant BCS-1748285 and the Center for Health and Wellbeing grant to A.C.

## Author contributions

I.M. and A.C. were involved in designing the research and analyzing the data and A.D. was involved in collecting and coding the data. In addition, I.M., A.D., and A.C. wrote the paper.

## Additional information

**Competing interests:** The authors declare no competing interests.

**Journal Peer Review Information:** *Nature Communications* thanks the anonymous reviewer(s) for their contribution to the peer review of this work. Peer reviewer reports are available.

