## [Peer Review File · Nature Communications]

Editorial Note: Reviewer's name has been replaced with "the reviewer" to ensure confidentiality.

Reviewers' comments:

Reviewer #1 (Remarks to the Author):

This paper focuses on the emergence of shared memories in social networks. The main strengths of the paper are its empirical nature, with the design of a convincing framework to test their hypothesis, and a study of the impact of the order of temporal interactions on the convergence of the dynamics. The impact of modularity on convergence is well-established in standard classical models (e.g. spectral gap for linear models) and experiments, but the focus on the temporal aspect of the conversations is, I believe, a novel and important contribution. The results are explained by qualitative, intuitive arguments, and is expected to trigger theoretical research, for instance by means of threshold models, in the temporal networks community. Besides these theoretical implications, the message is also important and impactful for researchers in the social sciences. For these reasons, it is without hesitation that I recommend this work for publication in Nature Communications.

As a minor comment, the authors could be interested in:

Bursty communication patterns facilitate spreading in a threshold-based epidemic dynamics, by Takaguchi, Taro and Masuda, Naoki and Holme, Petter, PloS one considering, by means of numerical simulations, a somehow related problem.

Reviewer #2 (Remarks to the Author):

In this manuscript, the authors report an experiment designed to test the effects of temporal sequence in conversations within a social network on the emergence of shared memories. The authors build on their recent work where they used a 10-person network (Coman, et al., 2016) and reported micro-level dyadic interaction effects on memory giving rise to collective memory at the network level. In the present experiment, the authors employed a 16-person network methodology to test the role of temporal sequence. All participants studied a short narrative/story individually and then performed three sequential recall tasks. The first and third recalls were carried out individually (pre- conversational and post-conversational recalls, respectively). The second recall occurred between dyads within the 16-person network where the dyads worked on a computer-mediated platform. Each member of a dyad was seated at a visually partitioned, computer terminal to type their own responses, and performed dyadic recall with their partner by each taking turn to type item units from the story.

The authors' aim was to test whether the sequence of joint recalls that occur within-clusters first (cluster-ties) versus across-clusters first (bridge-ties) changes memory convergence. Among the key findings the authors report, dyadic recall in the Bridge Ties First condition led to greater memory convergence than in Clusters Ties First condition. The mnemonic similarity scores were also greater in the Bridge Ties First condition than in the Cluster Ties First condition for participants who were part of the same cluster or in neighboring clusters but not if they were in distant clusters. Finally, the authors also report that less within-cluster similarity was observed in the Cluster Ties First condition compared to the Bridge Ties First condition.

The manuscript addresses interesting questions about the process by which collective memory emerges, and reports several analyses to report micro-level and macro-level memory similarity/convergence as a function of the temporal sequence of dyadic recall. This question is interdisciplinary and the authors draw on the ideas from network theory and from behavioral cognitive experiments to map the emergence of collective memory. The group-level methodology used in this study offers several possibilities for examining the process of memory convergence. I enjoyed reading the Introduction that situated this work within a broader context. The main question about how temporal sequence of conversations influences formation of collective memory is important and interesting, the specific questions about how the bridge ties influence collective memory are well developed, and the experimental test of these questions is informative. The use of accumulated overlap measure is particularly useful and revealing. I am enthusiastic about the aims of this work and the network methodology used here to test novel questions about mapping the emergence of collective memory.

There are four points in this version of the manuscript that I would like the authors to consider further:

1) Several findings reported in this manuscript contribute to our understanding of the nature of mnemonic similarity and convergence at the dyadic level and at the network level. Among these, two findings just meet or come close to the traditional p value threshold (Figure 4 caption - An important prediction that the Bridge-ties first condition would yield greater memory convergence than the Cluster-ties first condition was observed at $p = .052$; another important prediction of greater mnemonic similarity for Bridge-ties First than Cluster-ties first participants in neighboring clusters was observed at $p = .06$). For these (and other) findings, in addition to the confidence intervals the authors already report additional information about effect size and power/ sample size, as appropriate, may be added. The authors could draw upon their prior work (Coman et al., 2016) to assess the latter where they used similar measures of dyadic-level and network-level memory emergence.

2) The manuscript develops useful references to network theory by bringing in concepts of topology, centrality, and links. In addition, there are connections to another related literature that are important to draw in this work. This line of work consists of behavioral experiments on conversational remembering in groups and is related to the questions tested here. Since the reported study is itself a cognitive-behavioral experiment conducted with human participants, in my view it is especially relevant to include references to this related literature. Here are some concrete examples. A goal of the reported work is to test the relationship between "micro-level cognitive processes and the structural and temporal features of social networks" that "can give rise to large-scale social phenomena" (page 3). Further, in the General Discussion, the authors note the importance of investigating the impact of network size and structure on collective memory development. Experiments have that tested related aims. Researchers have developed a framework for (Rajaram & Pereira-Pasarin, 2010, *Perspectives in Psychological Science*), and reported findings from experimental investigations about, the contributions of several micro-level cognitive processes including retrieval disruption, re-exposure, reminiscence, and forgetting that characterize the collaborative remembering processes and influence the formation of collective memory. Researchers have also tested these questions by varying group/network structures consisting of insular (or identical) groups versus diverse (or reconfigured) groups and using group structures that were composed of three or nine members (Choi, Blumen, Congleton, & Rajaram 2014, *Journal of Cognitive Psychology*; Choi, Kensinger, & Rajaram, 2017, *Journal of Experimental Psychology: General*).

In the General Discussion, the authors note that community convergence for inaccurate information can arise from many sources, including "slightly different events across the community". Related to this idea, there's research using an experimental design where information was distributed differently across different members within a group. Some information was seen by all, other information by some, and yet other information by only one member within the group. This design created an opportunity to observe possible false memory effects at the level of the collective, and findings showed that false collective memory was higher in identical (insular) groups than in diverse groups (Choi et al., 2017). In these studies, the observed patterns of higher collective forgetting (Choi et al., 2014) and higher false collective memory (Choi et al., 2017) in insular network structures connect with the idea of "information bubbles" discussed in the present manuscript. Also, interactions with different conversational partners in the reconfigured groups in experiments increased the diversity of remembered information compared to insular groups, similar to the idea of "information diversity" discussed in the present manuscript. In addition, the authors report an intriguing finding where later rounds of conversation disrupt the alignment of memories achieved in earlier rounds, that is, the authors found less within-cluster similarity in the Cluster Ties First condition compared to Bridge Ties First condition. This might seem like a counterintuitive finding but a similar influence has been shown in work with repeated collaborative recalls, where later collaboration exerted greater influence than earlier collaboration on subsequent individual memory structures (Congleton & Rajaram, 2014, *Journal of Experimental Psychology: General*; Choi et al., 2014).

For the reasons outlined above, the noted articles are highly related and should be cited.

3) A discussion about the manner of conversational recall would strengthen the report of the procedure used. Some past studies have used face-to-face and interactive procedures that are more conversational than the computer-mediated platform used here. The practice of turn-taking for recall used here is also frequently reported in published experiments on conversational recall. However, in this study participants were seated separately at visually partitioned computers, they did not know the identity of their dyadic partner, and each participant anonymously entered into the computer their recall items. As such, a discussion about whether the terminology "conversational recall" is entirely apt to describe this procedure would be particularly useful for future investigations.

4) The authors might consider reducing the number of different terminologies used to describe collective memory unless these are absolutely necessary. Presently, the following terms are used interchangeably, "community-wide synchronization of memories", "mnemonic convergence", "collective memory", "shared remembering", "shared memory, and "degrees of convergence". To an uninitiated reader, it can get difficult to hold so many terms in mind while also distinguishing these terms from the computational terminologies used (e.g., "mnemonic similarity" and mnemonic convergence", which have distinct and clearly operationalized meanings in this study).

Other points -

- 1) What was the duration of the pre-conversational and post-conversational recall phases?
- 2) It would be helpful to know whether during dyadic recall each member of the dyad was able to see on the computer monitor the recall units typed by their partners or whether each member saw only their own recalled items. This detail is important for understanding the nature of collaboration during recall.
- 3) It is noted that the recall distortion was low and that the recall scoring system was binary.

Did participants produce any recall distortions (false constructions but traceable to some details in the story) as commonly found in narrative recall, and if they did, how were these handled?

4) The reference list and the cited articles do not match fully. For example, articles 31, 32, 33, 37, 38, and 39 are included in the Reference list but not cited within the manuscript.

Reviewer #3 (Remarks to the Author):

Collective memory is a fascinating topic and I applaud the interest in studying it experimentally. Overall, it's a very interesting project. I have a few questions about it, and a few suggestions that I hope will be helpful. Most of my thoughts focus on what I'd like to see the authors do next with this work after this paper.

1) The motivating description of the phenomenon of large scale, community wide convergence on shared memories (or beliefs) seems a little far from the kind of setup this study uses. It's not that the study needs to be much larger in size, as much as that the network architecture that is used is somewhat in the tradition of stylized small-group research. I'm really curious to know what would happen if there were enough people (even just twenty-six), so that the network were more modular (with well-defined clusters, with variable number of bridge ties across them).

2) It would be cool to have different conditions in which the authors vary the width of the bridges across the clusters, to see at what point the order of interactions no longer matters for the outcome. If there was a sharp transition in the collective memory dynamics, this would help to identify something that might be called the 'critical bridge width', which would be the point at which the order of interactions no longer matters for the process of integrating collective memories across groups. See for instance Centola's discussion in his book about the spread of behavior in social networks.

3) I am interested to know how the results on memory diversity would play out if the topics were either more emotionally poignant (like topics related to war), or more related to an issue where people engage in some degree of motivated reasoning (like a politically contentious topic). In that case, this framework for studying collective memory I think would be quite important because it would then help us to understand the formation of long held cultural biases, and the ways that networks help in the formation or dissolution of those biases. I believe something like this is in the background with this study, and I'd be very happy to see more work that explicitly brings out these topics.

Overall, I think the authors have done a nice job with this paper. I would be happy to see this study published, and for the authors to build on this work to pursue some of the topics I have described above.

Reviewer #1 (Remarks to the Author):

1. This paper focuses on the emergence of shared memories in social networks. The main strengths of the paper are its empirical nature, with the design of a convincing framework to test their hypothesis, and a study of the impact of the order of temporal interactions on the convergence of the dynamics. The impact of modularity on convergence is well-established in standard classical models (e.g. spectral gap for linear models) and experiments, but the focus on the temporal aspect of the conversations is, I believe, a novel and important contribution. The results are explained by qualitative, intuitive arguments, and is expected to trigger theoretical research, for instance by means of threshold models, in the temporal networks community. Besides these theoretical implications, the message is also important and impactful for researchers in the social sciences. For these reasons, it is without hesitation that I recommend this work for publication in Nature Communications.

As a minor comment, the authors could be interested in "Bursty communication patterns facilitate spreading in a threshold-based epidemic dynamics," by Takaguchi, Taro and Masuda, Naoki and Holme, Petter, PloS one considering, by means of numerical simulations, a somehow related problem.

We thank the reviewer for highlighting the strengths of our paper and for contextualizing our findings more generally for the social sciences. The temporal nature of network interactions has been, in our view, an insufficiently explored research topic in network science. We now added the reference on threshold models suggested by the reviewer; we believe that it is indeed relevant in the context of our investigation.

Reviewer #2 (Remarks to the Author):

The manuscript addresses interesting questions about the process by which collective memory emerges, and reports several analyses to report micro-level and macro-level memory similarity/convergence as a function of the temporal sequence of dyadic recall. This question is interdisciplinary and the authors draw on the ideas from network theory and from behavioral cognitive experiments to map the emergence of collective memory. The group-level methodology used in this study offers several possibilities for examining the process of memory convergence. I enjoyed reading the Introduction that situated this work within a broader context. The main question about how temporal sequence of conversations influences formation of collective memory is important and interesting, the specific questions about how the bridge ties influence collective memory are well developed, and the experimental test of these questions is informative. The use of accumulated overlap measure is particularly useful and revealing. I am enthusiastic about the aims of this work and the network methodology used here to test novel questions about mapping the emergence of collective memory.

We thank the reviewer for the assessment of our work given her expertise on collaborative remembering and collective memory. We address her more specific comments below, noting that we integrated this feedback in the revised manuscript

and we believe the paper improved as a consequence.

There are four points in this version of the manuscript that I would like the authors to consider further:

1) Several findings reported in this manuscript contribute to our understanding of the nature of mnemonic similarity and convergence at the dyadic level and at the network level. Among these, two findings just meet or come close to the traditional p value threshold (Figure 4 caption - An important prediction that the Bridge-ties first condition would yield greater memory convergence than the Cluster-ties first condition was observed at $p = .052$; another important prediction of greater mnemonic similarity for Bridge-ties First than Cluster-ties first participants in neighboring clusters was observed at $p = .06$). For these (and other) findings, in addition to the confidence intervals the authors already report additional information about effect size and power/ sample size, as appropriate, may be added. The authors could draw upon their prior work (Coman et al., 2016) to assess the latter where they used similar measures of dyadic-level and network-level memory emergence.

We followed the reviewer's suggestions and supplemented the main statistical analyses with effect size estimates and statistical power computations. We note that even though the two t-test comparisons only reached marginal significance, the effect size computations indicate large and reliable effects (all critical comparisons have an effect size, estimated using Cohen's d, larger than 1). The reason for these large effect sizes is the aggregation procedure, which involves averaging over 120 pairwise comparison scores to compute a single mnemonic convergence score in each network. This aggregation procedure drastically reduces the standard deviation and standard error in each condition, therefore increasing the likelihood of observing an effect if one exists. We now make this clear on the bottom of page 19:

"The stopping rule for participant recruitment was established based on the effect size obtained in a previous study that used aggregate measures of collective memory in networks (Coman et al., 2016). A sample size of 12 networks was deemed adequate to obtain an effect size of $Cohen's d = 1$ for the planned between-condition comparisons. The relatively small sample size needed to reveal an effect is due to the fact that the aggregation procedure involves averaging over all pairwise scores within each network (i.e., 120 scores per network), which drastically reduces the standard deviation and standard error in each condition."

2) The manuscript develops useful references to network theory by bringing in concepts of topology, centrality, and links. In addition, there are connections to another related literature that are important to draw in this work. This line of work consists of behavioral experiments on conversational remembering in groups and is related to the questions tested here. Since the reported study is itself a cognitive-behavioral experiment conducted with human participants, in my view it is especially relevant to include references to this related literature. Here are some concrete examples. A goal of the reported work is to test the

relationship between "micro-level cognitive processes and the structural and temporal features of social networks" that "can give rise to large-scale social phenomena" (page 3). Further, in the General Discussion, the authors note the importance of investigating the impact of network size and structure on collective memory development. Experiments have tested related aims. Researchers have developed a framework for (Rajaram & Pereira-Pasarin, 2010, *Perspectives in Psychological Science*), and reported findings from experimental investigations about, the contributions of several micro-level cognitive processes including retrieval disruption, re-exposure, reminiscence, and forgetting that characterize the collaborative remembering processes and influence the formation of collective memory. Researchers have also tested these questions by varying group/network structures consisting of insular (or identical) groups versus diverse (or reconfigured) groups and using group structures that were composed of three or nine members (Choi, Blumen, Congleton, & Rajaram 2014. *Journal of Cognitive Psychology*; Choi, Kensinger, & Rajaram, 2017, *Journal of Experimental Psychology: General*).

In the General Discussion, the authors note that community convergence for inaccurate information can arise from many sources, including "slightly different events across the community". Related to this idea, there's research using an experimental design where information was distributed differently across different members within a group. Some information was seen by all, other information by some, and yet other information by only one member within the group. This design created an opportunity to observe possible false memory effects at the level of the collective, and findings showed that false collective memory was higher in identical (insular) groups than in diverse groups (Choi et al., 2017). In these studies, the observed patterns of higher collective forgetting (Choi et al., 2014) and higher false collective memory (Choi et al., 2017) in insular network structures connect with the idea of "information bubbles" discussed in the present manuscript. Also, interactions with different conversational partners in the reconfigured groups in experiments increased the diversity of remembered information compared to insular groups, similar to the idea of "information diversity" discussed in the present manuscript. In addition, the authors report an intriguing finding where later rounds of conversation disrupt the alignment of memories achieved in earlier rounds, that is, the authors found less within-cluster similarity in the Cluster Ties First condition compared to Bridge Ties First condition. This might seem like a counterintuitive finding but a similar influence has been shown in work with repeated collaborative recalls, where later collaboration exerted greater influence than earlier collaboration on subsequent individual memory structures (Congleton & Rajaram, 2014, *Journal of Experimental Psychology: General*; Choi et al., 2014).

For the reasons outlined above, the noted articles are highly related and should be cited.

We thank the reviewer for pointing us to these references. In addition to the previously cited references we integrated several other suggestions as part of our argument. More specifically, we now cite Rajaram & Pereira-Pasarin (2010) for a framework aimed at bridging between micro-level cognitive phenomena and collective memory. We also cite Choi, Kensinger, & Rajaram (2017) in the discussion section as evidence of empirical work aimed at investigating the

formation of false collective memories. Finally, in support of our finding that the within-cluster average similarity was larger in the Bridge Ties First condition than in the Cluster Ties First condition we now cite Congleton & Rajaram (2014).

3) A discussion about the manner of conversational recall would strengthen the report of the procedure used. Some past studies have used face-to-face and interactive procedures that are more conversational than the computer-mediated platform used here. The practice of turn-taking for recall used here is also frequently reported in published experiments on conversational recall. However, in this study participants were seated separately at visually partitioned computers, they did not know the identity of their dyadic partner, and each participant anonymously entered into the computer their recall items. As such, a discussion about whether the terminology "conversational recall" is entirely apt to describe this procedure would be particularly useful for future investigations.

The reviewer raises an interesting point. We do believe that participants engaged in conversations, despite the fact that they didn't know one another's identity. We purposefully did not provide identifying information to the interaction partners, so that we could insulate the phenomena under investigation from the complexity of the rich social cuing that occurs during face-to-face conversations. We do agree with the reviewer, though, that contrasting face-to-face conversations with computer-mediated interactions will likely reveal meaningful dynamics. We now discuss such research trajectories, on the bottom of page 16:

"For instance, in our current investigation participants interacted with one another in computer-mediated interactions that involved turn-taking. Even though the experimental situation we created is representative of today's social media landscape (e.g., Twitter, Facebook), it is likely that face-to-face conversations in which social cues are much more salient would result in different outcomes. More research is needed, thus, into how face-to-face conversations might affect large-scale phenomena differently than computer-mediated interactions. Our conjecture is that the more social information the participants have of one another, the more pronounced the social influence processes will be. More specifically, similarity cues would facilitate social influence, while dissimilarity cues would attenuate social influence, relative to control conditions in which little or no social information is provided."

4) The authors might consider reducing the number of different terminologies used to describe collective memory unless these are absolutely necessary. Presently, the following terms are used interchangeably, "community-wide synchronization of memories", "mnemonic convergence", "collective memory", "shared remembering", "shared memory, and "degrees of convergence". To an uninitiated reader, it can get difficult to hold so many terms in mind while also distinguishing these terms from the computational terminologies used (e.g., "mnemonic similarity" and mnemonic

convergence", which have distinct and clearly operationalized meanings in this study).

We agree that the terminology we used might have detracted from comprehension. To address this concern, we undertook two measures: (1) we referred to the "formation of collective memories" in the introduction and discussion section (and discarded "community-wide synchronization," "mnemonic convergence," and "shared memory"), and (2) we clarified that we operationalized the formation of collective memories as "mnemonic convergence," in the middle of page 8:

"For our dependent measures, we computed scores that quantitatively captured the formation of collective memory in the community. We refer to these scores as mnemonic convergence scores when they involve the entire community (of 16 participants) and mnemonic similarity scores when they involve only sub-sections of the community."

We believe that the simplification of the terminology improved the readability of the paper.

Other points -

1) What was the duration of the pre-conversational and post-conversational recall phases?

The participants had as much time as needed to type their recalls in the pre- and post-conversational recall phases. Looking at the timing of these recalls in our data, they ranged between 4 and 8 minutes. We now report this on the bottom of page 21.

2) It would be helpful to know whether during dyadic recall each member of the dyad was able to see on the computer monitor the recall units typed by their partners or whether each member saw only their own recalled items. This detail is important for understanding the nature of collaboration during recall.

The issue brought up by the reviewer here is related to the previous point as to whether the interactions that participants engaged in could be labeled as conversational recalls. Indeed, the participants chatted with one another in interactive exchanges and they could see the entries of the other participant on the screen as they occurred and respond to them in real time. We now clarify this on page 21:

"Of note, both participants who engaged in a conversation saw the conversation as it unfolded on the screen and were able to scroll to any point of the conversation they wished."

3) It is noted that the recall distortion was low and that the recall scoring system was binary. Did participants produce any recall distortions (false constructions but traceable to some details in the story) as commonly found in narrative recall, and if they did, how were these handled?

To code the data, we followed the coding scheme developed by Anderson & Pichert (1978), as reported in the Methods section and the Supplementary Material 1. This coding scheme resulted in very few recall distortions because of the gist-based coding recommended in the coding manual. For instance, if a participant recalled that: "there were two bikes in the garage," the unit was counted as remembered (despite the factual inaccuracy, since there were 3 bikes mentioned in the studied story). We coded it this way because in the coding manual the instructions indicated that remembering "there were bikes in the garage" should be counted as correctly recalled.

Due to this gist-based coding, we had very few recall distortions in the data (<1%) and we coded these distortions in a gist-based consistent manner, with recall units that contained egregious errors not being counted as recalled (e.g., "there was a TV" when in fact "there was a stereo") and minimal errors were counted as correctly remembered (e.g., "there was a computer" instead of the studied item: "there was a laptop").

To make this clear, we added, on page 22: "The coding scheme followed the designation employed in the original study by (16) for 30 predefined memory items included in the story (see *Supplementary Materials 1*). For each item, a score of 1 indicated that it was remembered and 0 indicated that the item was not recalled. Items were coded for "gist" recall, meaning that the participant did not need to recall each scoring unit verbatim in order for it to count. For example, although a scoring unit on the scale reads "The basement is damp", if a participant mentioned a "soggy cellar," the unit was counted as remembered. Due to this gist-based coding, we had very few recall distortions in the data (<1%) and we coded these distortions in a gist-based consistent manner, with recall units that contained egregious errors not being counted as recalled (e.g., "there was a TV" when in fact "there was a stereo") and minimal errors were counted as correctly remembered (e.g., "there was a computer" instead of the studied item: "there was a laptop")."

4) The reference list and the cited articles do not match fully. For example, articles 31, 32, 33, 37, 38, and 39 are included in the Reference list but not cited within the manuscript.

We thank the reviewer for pointing this inconsistency. It is due to the different drafts that were circulated among the authors of the paper. We now addressed it by citing references 31 (Congleton et al.), 32 (Marsh et al.), 37 (Salganik et al.), and 38 (Masuda et al.) in text, and discarding 33 (Ebbinghaus et al.) and 39 (Bauml et al.), which are no

longer relevant for the argument we present in the paper.

Reviewer #3 (Remarks to the Author):

Collective memory is a fascinating topic and I applaud the interest in studying it experimentally. Overall, it's a very interesting project. I have a few questions about it, and a few suggestions that I hope will be helpful. Most of my thoughts focus on what I'd like to see the authors do next with this work after this paper.

We thank the reviewer for this assessment and we respond to their specific comments below. We are actively conducting research in the directions identified by the reviewer and we made these trajectories clearer in the discussion section, as described below.

1) The motivating description of the phenomenon of large scale, community wide convergence on shared memories (or beliefs) seems a little far from the kind of setup this study uses. It's not that the study needs to be much larger in size, as much as that the network architecture that is used is somewhat in the tradition of stylized small-group research. I'm really curious to know what would happen if there were enough people (even just twenty-six), so that the network were more modular (with well-defined clusters, with variable number of bridge ties across them).

Our reasoning is very much in line with the reviewer's thinking on both indicated directions. First, on the variable number of bridge ties, we have one paper under review that uses 16-member networks, just as the current paper, but employs different numbers of bridge ties between clusters. The results of that paper are very much in line with the ones reported here; in addition, for that project, we also look at how an individual's network position impacts the emerging collective memory of the community. In addition, we have an ongoing NSF-funded study that expands the communities to 20-member networks and creates propagation chains to understand how the degree of propagation of certain cognitive effects impacts the formation of collective memories. We are still collecting data for this latter project, but we expect cognitive effects that have larger degrees of propagation (i.e., they propagate deeper in the chain) to impact the emerging collective memory of the community to a larger extent. We now make these points clearer in the middle of page 17:

"At the same time, establishing whether social influences on memory involve simple contagions (one source necessary for memory change) or complex contagions (multiple sources necessary for memory change) will likely impact the types of networks most conducive to the formation of collective memories or the propagation and maintenance of misinformation within a community (1, 13, 45). In this context, varying the number of bridge ties between clusters would provide more nuanced answers to questions about the optimal connectivity among clusters that would facilitate the community-wide convergence on the same memory about a past event."

2) It would be cool to have different conditions in which the authors vary the width of the bridges across the clusters, to see at what point the order of interactions no longer matters for the outcome. If there was a sharp transition in the collective memory dynamics, this would help to identify something that might be called the 'critical bridge width', which would be the point at which the order of interactions no longer matters for the process of integrating collective memories across groups. See for instance Centola's discussion in his book about the spread of behavior in social networks.

This is an interesting suggestion. Contrasting bridge ties first with cluster ties first as we manipulate the bridge width would not be as informative, since in the cluster ties first condition information would simply remain within the cluster without affecting neighboring clusters. We agree with the reviewer that manipulating the width of the bridge ties (in the bridge ties first condition only) would reveal the critical bridge width at which the community no longer reaches a significant degree of community-wide convergence.

We thank the reviewer for the suggestion and we're certainly interested in pursuing such an idea. We note that in some of our previous work, mnemonic influence propagated no more than three degrees from the originating source, so we speculate that the critical bridge width should be somewhere between one and three. Following the reviewer's suggestion, we now state, on the top of page 17:

"At a network level, one could investigate what is the critical bridge width (i.e., the length of the chain that bridges between two clusters) at which collective memories are still affected by the temporal sequencing of conversations. This investigation is critical, given that in real-world communities, bridges among clusters are oftentimes of widths larger than 1. Previous investigations suggest that the social influence from one individual to another does not spread more than 3 degrees away from the originating source, which might constitute the upper limit at which the critical bridge width can still affect mnemonic convergence."

3) I am interested to know how the results on memory diversity would play out if the topics were either more emotionally poignant (like topics related to war), or more related to an issue where people engage in some degree of motivated reasoning (like a politically contentious topic). In that case, this framework for studying collective memory I think would be quite important because it would then help us to understand the formation of long held cultural biases, and the ways that networks help in the formation or dissolution of those biases. I believe something like this is in the background with this study, and I'd be very happy to see more work that explicitly brings out these topics.

We again thank the reviewer for a great suggestion. We have now added these suggestions to the discussion section and we note that one of the studies we're conducting at the moment involves creating an ingroup/outgroup manipulation and

using more group-relevant materials to understand how these manipulation impacts the propagation of mnemonic influence and facilitates/attenuates convergence. We also cite a recent paper from Damon Centola's group that was published a few days ago on the impact of partisan (political) bias on social influence in networks. We now elaborate on these issues in the middle of page 18:

"We focused on how communities come to hold a collective memory of an emotionally- neutral event that all participants experienced. We made the decision to restrict the experimental situation to this setting in order to rigorously isolate the variables that are of interest to this investigation. In real world-contexts, however, people often encounter emotionally-charged events, the propagation of which might be modulated by motivated reasoning processes. Based on previous research, we conjecture that emotion should facilitate the propagation of social influence in networks in a way that would increase community-wide convergence (38). If one experiences an emotional reaction when witnessing an event, then one is more likely to relay it to another individual, which in turn would facilitate convergence processes (39). This propagation- convergence association would be influenced, however, by people's prior beliefs, memories, and motivations. That is, groups of similar-minded individuals might perceive an event through the prism of their identity in a way that makes them converge on a similar memory of the past. These motivational biases could lead to different rival groups to converge on different memories of the same event, as famously shown in previous research (40, 41). Oftentimes, these biases create a context for communities to converge on inaccurate information (42). In the current investigation, the rate of memory distortion was very low, which limits the conclusions one could draw based on this data. But systematic investigations involving: (i) slightly different events across the community, (ii) manipulating the ambiguity of the experienced event, and (iii) the motivations of the community members, will likely reveal meaningful dynamics involving the formation of false collectivememories."

Overall, I think the authors have done a nice job with this paper. I would be happy to see this study published, and for the authors to build on this work to pursue some of the topics I have described above.

We once again thank the reviewers for their thoughtful and constructive comments and for their positive assessment of our research. We hope that the current version of the manuscript addressed their comments and integrated their suggestions to a satisfactory degree. We certainly feel that the paper has greatly improved as a result of these changes.

****REVIEWERS' COMMENTS**

Reviewer #1 (Remarks to the Author):

The authors have addressed my minor comments. My recommendation is an accept.